# Quality of maternal and newborn care in limited-resource settings: a facility-based cross-sectional study in Burkina Faso and Côte d'Ivoire

Tieba Millogo  ,[1,2] Marie Laurette Agbre-Yace,[3] Raissa K Kourouma,[3] W Maurice E Yaméogo,[2] Akoua Tano-Kamelan,[3] Fatou Bintou Sissoko,[2] Aminata Soltié Koné-Coulibaly,[3] Anna Thorson,[4] Seni Kouanda[1,2]

¹Epidemiology, African Institute of Public Health, Ouagadougou, Kadiogo, Burkina Faso
²Biomedicine and Public Health, Institut de Recherche en Sciences de la Sante, Ouagadougou, Centre, Burkina Faso
³Institut National de Santé Publique (INSP-Côte d'Ivoire), Abidjan, Côte d'Ivoire
⁴Human Reproduction Program/ World Health Organization (Geneva), Geneva, Switzerland

**Correspondence to**
Dr Tieba Millogo;
millogorod@gmail.com

## ABSTRACT

**Objective** To assess and compare the quality of intrapartum and immediate postpartum care across levels of healthcare in Burkina Faso and Côte d'Ivoire using validated process indicators.

**Design** Health facility-based cross-sectional study with direct observation of healthcare workers' practices while caring for mother–newborn pairs during intrapartum and immediate postpartum periods.

**Setting** Primary healthcare facilities and their corresponding referral hospitals in the Central-North region in Burkina Faso and the Agneby-Tiassa-Mé region in Côte d'Ivoire.

**Participants** Healthcare providers who care for mother–newborn pairs during intrapartum and immediate postpartum periods, the labouring women and their newborns after childbirth.

**Main outcome measure(s)** Adherence to essential best practices (EBPs) at four pause points in each birth event and the overall quality score based on the level of adherence to the set of EBPs observed for a selected pause point.

**Results** A total of 532 and 627 labouring women were included in Burkina Faso and Côte d'Ivoire, respectively. Overall, the compliance with EBPs was insufficient at all the four pause points, even though it varied widely from one EBP to another. The adherence was very low with respect to hand hygiene practices: the care provider wore sterile gloves for vaginal examination in only 7.96% cases (95% CI 5.66% to 11.06%) in Burkina Faso and the care provider washed hands before examination in 6.71% cases (95% CI 3.94% to 11.20%) in Côte d'Ivoire. The adherence was very high with respect to thermal management of newborns in both countries (>90%). The overall mean quality scores were consistently higher in referral hospitals in Burkina Faso at all pause points excluding immediate post partum.

**Conclusions** Women delivering in healthcare facilities do not always receive proven EBPs needed to prevent poor childbirth outcomes. There is a need for quality improvement interventions.

## INTRODUCTION

Reducing the high burden of preventable maternal and neonatal deaths remains an important health-related goal in the sustainable development era. Although tremendous progress has been achieved globally in this area during the last two decades, most of the highly affected countries have failed to meet the target of reducing their 1990 maternal mortality ratios by 75% by 2015.[1 2] Around 830 women are still dying every day from pregnancy and childbirth-related complications, of which 99% are occurring in low-income and middle-income countries and more than 50% in sub-Saharan Africa (SSA).[1 3] There is strong evidence showing that the largest burden of maternal death is clustered around the time of childbirth and immediate postpartum period.[4] A global advocacy for skilled attendance at each birth, aiming to reduce preventable maternal deaths during the Millennium Development Goals era,[5 6] resulted in a significant increase

---

## Strengths and limitations of this study

► Random selection of health facilities in two different countries and observation of sufficient sample sizes of birth events in each country.
► Direct and prospective assessment of healthcare workers' adherence to essential best practices (EBPs) during and after childbirth by trained healthcare providers as opposed to retrospective ascertainment through medical records in many previous studies.
► Use of context-specific validated EBPs to maximise relevance of the study findings.
► Each country data are not representative of the whole country.
► Because of the direct observation of the practices, we cannot rule out the Hawthorne effect and the data were collected by eight data collectors in each country and are thus subject to an inter-rater reliability bias even though they were trained to standardise the processes.



in the access and uptake of childbirth services globally, including in SSA countries.[7 8] This was, however, not paralleled by the expected reduction in maternal and infant mortality rates in low-resource settings.[9] A skilled birth attendant and health facility delivery will improve maternal and newborn outcomes only if they encompass the implementation of proven effective interventions needed to prevent or manage major causes of maternal and newborn morbidity and mortality.

When the quality of care is substandard, these principles are challenged.[10] The poor quality of care in health facilities, specifically in low-resource settings, represents a neglected agenda[11] and is increasingly recognised as a major contributing factor to persistent high levels of preventable maternal and newborn morbidity and mortality.[10] Closing the gaps in quality of care will be central to the achievement of the global commitment towards Universal Health Coverage (UHC).[12] One of the pillars of the UHC will be to make quality services available to individuals and communities to meet their health-related needs without experiencing financial hardships. Quality of care is however complex and encompasses various approaches and dimensions. Quality of maternal and newborn care is very often assessed through delivery outcomes or the availability of resources needed to implement effective interventions,[13–15] rarely using process indicators.[16] While childbirth outcomes are relevant indicators of quality of care, they do not provide information on the process of care, and therefore, neither on quality gaps that need to be filled. Very few studies have investigated the quality of maternal and newborn care in SSA using process indicators. In this study, we intend to assess and compare the quality of intrapartum and immediate postpartum care in two SSA countries using established process indicators.[16]

## METHODS
### Study settings
Burkina Faso (low-income country) and Côte d'Ivoire (low-middle income country), two SSA countries with maternal mortality ratios among the highest in the world (310 and 614 per 100000 live births, respectively),[17 18] have both witnessed significant increase in the uptake of childbirth services during the last two decades, with health facility births reaching nearly 90%.[19] Yet, both countries are struggling to achieve the desire reduction in maternal deaths and the current pace must be further accelerated if they are to meet Target 3.8 of the Sustainable Development Goals of not having more than 140 maternal deaths per 100000 live births by 2030.[1] The study was carried out in two regions where the collaborating research centres have locations, one in each country. In Burkina Faso, the Central-North health region hosted the study, while in Côte d'Ivoire, it was conducted in the Agneby-Tiassa-Mé health region. The two regions were purposively selected based on the presence of a geographical location of the research centres that partner to conduct the study. In

each region, we randomly drew five from the existing (six in each country).

Both regions have their capital cities located approximately 100 km away from the respective capital city of its country. The two regions have similar healthcare delivery systems with a regional hospital, followed by district hospitals, each serving a health district composed of several primary healthcare centres. The central-north health region in Burkina Faso has 6 health districts and 130 primary healthcare facilities (PHCFs) and the Agnéby-Tiassa-Mé health region of Côte d'Ivoire is divided into 7 health districts with 131 PHCFs.

### Study design
We used a facility-based cross-sectional study in government-owned health facilities, enabling a direct and passive observation of the healthcare providers' management of deliveries.

### Study population, sampling and sample size calculation
In each selected region, we randomly selected five health districts. The district hospitals of the selected health districts along with the regional hospital were systematically included in the study in each country. Based on the average childbirth caseloads per level of care (hospital and PHCF) and a maximum length of facility stay of 5 days per data collector's team, it was estimated that a minimum of 65 PHCFs would be needed in each country to achieve the calculated sample size. In the catchment area of each selected health district, up to 15 eligible PHCFs were randomly selected to be included in the study. Eligible PHCFs were those with a minimum average caseload of 1.5 childbirths per day as reported in the health records of the same period in the previous year (during the last quarter of 2017).

The study population comprised
► All the healthcare providers involved in intrapartum and immediate postpartum care, irrespective of their qualifications. They were offered to participate in the study and we excluded healthcare providers who were in the health facility for their qualifying internship.
► All the pregnant women in labour, admitted without c-section or referral needed to be identified at admission. We consecutively recruited all eligible pregnant women during the data collection period in each health facility. Exclusion criteria were being aged <18 years in both countries, pregnant women admitted at an advanced stage of the labour (third stage) with frequent (≤5 min apart) and intense uterine contractions that would prevent them from providing valid informed consent.

The sample size of pregnant women was calculated using the conservative proportion of 50% adherence to each essential best practice (EBP) with a precision of 5% and type I error of 5%. To account for the clustering nature of the data, the calculated sample size was adjusted using a design effect based on an intraclass correlation coefficient of 0.01[20] and a size of 15 observations per

group. The minimum sample size was estimated to be 425 pregnant women to be included in the observation per pause point in each country, giving a minimum total number of 1700 of observations per country.

## Data collection tools and processes

Data collectors with a medical background (medical students, midwives) and who had undergone a 2-day training including in study room role plays and a field testing of the study tools, were appointed in the delivery room to directly observe and document healthcare providers' adherence to EBPs at four critical points: admission, before the woman starts pushing, immediately post partum (soon after birth up to ≤1 hour), and before discharge from the health facility. During the field test, a pair of data collectors had to observe at least two pause points of the same birth event along with a supervisor. Data collectors' measurements were compared within each pair of data collectors and to the measurements of the supervisor. The observed variations were discussed.

The EBPs included in this study were derived from WHO Safe Childbirth Checklist (WHO-SCC)[21] complemented by a list of other validated quality indicators for SSA[16] (see table 1). A total of 34 EBPs were observed during each birth event. Adherence to each EBP was measured using a binary yes/no variable. The observation was strictly passive with minimum interaction between observer and healthcare workers and patients. The data collectors were only allowed to intervene in the event of a life-threatening complication. The data collection took place from 12 November to 20 December 2018 and lasted for up to 5 days in each health facility. Observations were conducted every day of the week and at any time without discontinuation, provided an eligible participant was present. A data collector could observe all or part of the four pause points for each birth event. However, data pertaining to the same critical juncture were fully collected by the same data collector. Thus, once a data collector had initiated the observation for a given pause

| No | Admission | Before pushing | Soon after birth | Before discharge |
|----|-----------|----------------|------------------|------------------|
| | **Table 1** Essential best practices to be observed for health workers' compliance at each pause point | | | |
| 1 | ► Washes his/her hands before examination | ► Sterile gloves available | ► Clamps or ties cord correctly | ► Assesses the mother for vaginal bleeding |
| 2 | ► Wears sterile gloves before vaginal examination | ► Soap and clean water available | ► Immediately dries the baby with clean towel | ► Assesses the baby for the quality of breast feeding |
| 3 | ► Asks for vaginal bleedings | ► Prepares a prefilled syringe with uterotonics for AMTSL | ► Assesses for vaginal and perineal lacerations | ► Discusses postpartum family planning with the woman/couple |
| 4 | ► Asks for vision blurred | ► Clean towel available | ► Assesses for placenta completeness | ► Plans postpartum visits and makes sure they will consult in the event of a complication |
| 5 | ► Checks HIV status | ► Sterile blade or scissor available | ► Assesses new born for need of reference | |
| 6 | ► Measures blood pressure | ► Face mask available | ► Takes mother vital signs within 15 min after | |
| 7 | ► Measures pulse | ► Suction device available | ► Palpates uterus within 15 min after birth | |
| 8 | ► Uses partograph where indicated | ► Assistant birth attendant identified and ready to- intervene | ► Initiates breastfeeding within 1 hour | |
| 9 | ► Measures temperature-yes | | ► Skin-to-skin contact initiated within 1 hour | |
| 10 | ► Assesses for need of referral | | ► Informs companion/woman to call if complications | |
| 11 | ► Encourages companion to assist | | | |
| 12 | ► Informs companion/woman to call in case of complications | | | |

AMTSL, active management of the third stage of labour.

point, he/she could get off work only after having filled in all the items pertaining to that pause point.

### Data analysis

Descriptive statistics were computed using means and SD or medians and IQRs for numerical variables, and proportions were used to describe categorical variables. We calculated the level of adherence to each EBP as the proportion of childbirth events in which the EBP was effectively performed. These proportions were computed along with their 95%CIs. For each pause point, based on complete cases, a quality score was computed as the number of EBPs effectively performed at each birth event. We further used the Wilcoxon rank sum test to compare quality scores at the four clinical junctures between PHCFs and hospitals in each country. Because the study finally included varied numbers of observations per health facility, all the analyses were weighted for the clinic size. A $p < 0.05$ indicated statistical significance.

Written informed consent was obtained from all participants of the study. Data were collected, managed and analysed in a way to ensure the confidentiality of study participants.

### Patient and public involvement

There has been no patient and/or public involvement in the study design, data collection, data analysis and writing of this research.

### RESULTS

A total of 73 and 69 health facilities in Burkina Faso and Côte d'Ivoire, respectively, were included in the study. In total, 532 and 627 pregnant women in labour gave informed consent to be included in the observations for pause-point I in Burkina Faso and Côte d'Ivoire, respectively. Once included in the observation (pause point I), a woman and thereafter her newborn were observed through the subsequent pause points unless they were referred to another health facility or transferred to a different service. Another reason why a mother–newborn pair could not be observed at the fourth pause-point was them still being in the health facility at the time the data collectors had completed their overall stay in that particular facility. The number of birth events observed by pause point by country and by health facility type are presented in table 2. As stated above, the reduction in participant numbers from each pause point to the next was a result of referral, transfer, and not yet being discharged from the health facility at the end of observation period.

### Compliance with EBPs at admission

In Burkina Faso, adherence to EBPs ranged from 7.96% (95% CI 5.66% to 11.06; wears sterile gloves for vaginal examination) to 99.27% (95% CI 97.74% to 99.76%; assesses the woman for need of referral). The adherence was significantly higher in referral hospitals as compared with PHCFs for the following EBPs: asks about vaginal bleeding ($p < 0.05$), asks about blurred vision ($p < 0.01$), measures blood pressure ($p < 0.001$), measures pulse ($p < 0.01$) and measures temperature ($p < 0.05$). Adherence was significantly higher in PHCFs for the EBP 'encourages companion/woman to call in case of complication' ($p < 0.01$).

In Côte d'Ivoire, adherence to EBPs ranged from 6.71% (95% CI 3.94% to 11.20%; washes his/her hands before examination) to 84.97% (95% CI 81.24% to 88.09%; assesses the woman for need of referral). The compliance was significantly higher in PHCFs than referral hospitals for the following EBPs: asks about vaginal bleeding ($p < 0.01$), measures blood pressure ($p < 0.001$), measures pulse ($p < 0.01$), measures temperature ($p < 0.01$) and encourages companion/woman to call in case of complication ($p < 0.001$).

### Compliance with EBPs at the immediate prepushing phase

In Burkina Faso, all EBPs were implemented in more than half of the deliveries that took place in referral hospitals. The same was observed in PHCFs, except for availability of bag and mask, which was observed in 39.34% (95% CI 34.38% to 44.53%) of the deliveries. The adherence was significantly higher in referral hospitals than PHCFs for three EBPs: prepares prefilled syringe with uterotonics for active management of the third stage of labour (AMTSL) ($p < 0.05$), bag and mask available ($p < 0.001$) and assistant birth attendant ready to intervene if needed ($p < 0.001$).

In Côte d'Ivoire, except for availability of bag and mask and availability of suction device, all other EBPs were adhered to in more than half of the deliveries in both types of health facilities. Bag and mask were available in 5.67% (95% CI 3.68% to 8.77%) and 18.74% (95% CI 12.17% to 27.69%) of the deliveries in PHCFs and referral hospitals, respectively. Availability of suction device ($p < 0.001$) and sterile blade or scissor ($p < 0.01$) were also better in referral hospitals than PHCFs and preparation of a prefilled syringe with uterotonics for AMTSL was best adhered to in PHCFs ($p < 0.001$).

### Compliance to EBPs immediately post partum

In Burkina Faso, adherence was above 50% for all EBPs ranging from 51.86% (95% CI 46.58% to 57.10%; takes vital signs within 15 min) to 99.71% (95% CI 98.67% to EBPs 99.94%; assesses the newborn for need of referral). Significantly higher adherence was observed in referral hospitals regarding encouraging companion/woman to call in case of complication ($p < 0.01$).

In Côte d'Ivoire, adherence ranged from 19.66% (95% CI 15.49% to 24.61%; clamps or ties cord correctly) to 98.5% (95% CI 96.91% to 99.28%; assesses for vaginal or perineal lacerations). The adherence was significantly higher in PHCFs for the following EBPs: Assesses for vaginal and perineal lacerations ($p < 0.01$), assesses newborn for need of referral ($p < 0.01$), palpates uterus within 15 min after birth ($p < 0.05$), initiates breastfeeding within 1 hour ($p < 0.01$) and skin-to-skin contact initiated within 1 hour ($p < 0.001$).

**Table 2** Compliance to essential best practices by pause point and by health facility type in Burkina Faso and Côte d'Ivoire

| Essential best practices | Burkina faso | | | | Côte d'Ivoire | | | |
|---|---|---|---|---|---|---|---|---|
| | PHCF | | RH | | PHCF | | RH | |
| | n/N | % (95% CI) | n/N | % (95% CI) | n /N | % (95% CI) | n/N | % (95% CI) |
| Admission | 428 | | 104 | | 513 | | 114 | |
| Washes his/her hands before examination | 86/420 | 20.50 (16.72 to 24.86) | 20/101 | 19.02 (12.48 to 27.89) | 29/501 | 06.72 (03.94 to 11.24) | 5/114 | 05.00 (02.10 to 11.45) |
| Wears sterile gloves before vaginal examination | 33/425 | 07.94 (05.64 to11.07) | 11/103 | 10.73 (06.03 to18.39) | 49/500 | 07.03 (05.23 to 09.39) | 6/114 | 05.60 (02.52 to 11.96) |
| Asks for vaginal bleedings | 181/422 | **41.62 (36.57 to 46.84)** | 55/104 | **54.18 (44.46 to 63.58)** | 144/504 | **33.97 (28.96 to 39.38)** | 23/114 | **20.40 (13.90 to 28.92)** |
| Asks for vision blurred | 160/422 | **37.04 (32.12 to 42.24)** | 53/104 | **52.51 (42.82 to 62.01)** | 122/504 | 27.67 (23.29 to 32.52) | 28/114 | 25.40 (18.12 to 34.38) |
| Checks HIV status | 333/425 | 79.25 (75.00 to 82.94) | 89/104 | 86.62 (78.85 to 91.83) | 344/497 | 66.08 (60.69 to 71.08) | 73/113 | 65.24 (55.94 to 73.51) |
| Measures blood pressure | 305/428 | **72.32 (67.48 to 76.68)** | 94/103 | **90.95 (83.45 to 95.25)** | 414/509 | **81.21 (76.66 to 85.04)** | 70/114 | **63.63 (54.42 to 71.94)** |
| Measures pulse | 170/428 | **39.87 (34.85 to 45.10)** | 54/102 | **54.36 (44.55 to 63.85)** | 230/506 | **48.57 (43.18 to 54.00)** | 35/114 | **33.56 (25.25 to 43.02)** |
| Uses partograph where indicated* | 225/428 | 53.11 (47.90 to 58.25) | 53/104 | 50.36 (40.73 to 59.97) | 90/508 | 17.72 (14.04 to 22.13) | 24/114 | 23.80 (16.57 to 32.94) |
| Measures temperature | 277/428 | **65.64 (60.50 to 70.43)** | 79/104 | **76.61 (67.49 to 83.79)** | 207/513 | **42.90 (37.60 to 48.36)** | 29/114 | **27.47 (18.83 to 36.70)** |
| Assesses for need of referral | 425/428 | **99.26 (97.74 to 99.76)** | 104/104 | 100 | 441/512 | **85.04 (81.27 to 88.16)** | 89/114 | **76.6 (67.65 to 83.67)** |
| Encourages companion to assist to the childbirth | 176/422 | **41.47 (36.49 to 46.63)** | 26/103 | **24.59 (17.22 to 33.82)** | 43/474 | 08.64 (06.22 to 11.88) | 11/109 | 11.14 (06.27 to 19.03) |
| Companion/woman informed to call if complication | 326/424 | 86.15 (82.06 to 89.42) | 101/104 | 96.7 (90.07 to 98.96) | 277/472 | **66.07 (60.93 to 70.86)** | 45/105 | **47.56 (38.03 to 57.26)** |
| Before pushing or before c-section | 416 | | 99 | | 477 | | 109 | |
| Sterile gloves available | 405/416 | 97.65 (95.78 to 98.71) | 96/99 | 96.70 (90.13 to 98.94) | 458/477 | 93.21 (87.48 to 96.43) | 104/109 | 95.54 (89.61 to 98.16) |
| Soap and clean water available | 259/416 | 63.42 (58.23 to 68.31) | 68/99 | 70.37 (60.73 to 78.49) | 365/477 | 71.15 (65.60 to 76.12) | 83/109 | 75.65 (66.56 to 82.91) |
| Prepares a prefilled syringe with uterotonics for AMTSL | 348/416 | **83.19 (79.05 to 86.65)** | 92/99 | **92.85 (85.71 to 96.56)** | 372/477 | **83.74 (80.22 to 86.74)** | 69/109 | **68.05 (58.93 to 75.96)** |

Continued

**Table 2** Continued

| Essential best practices | Burkina faso | | | | Côte d'Ivoire | | | |
|---|---|---|---|---|---|---|---|---|
| | PHCF | | RH | | PHCF | | RH | |
| | n/N | % (95% CI) | n/N | % (95% CI) | n /N | % (95% CI) | n/N | % (95% CI) |
| Clean towel available | 389/416 | 94.41 (91.93 to 96.17) | 96/99 | 96.90 (90.80 to 99.00) | 416/477 | 85.97 (82.23 to 89.02) | 102/109 | 92.92 (85.87 to 96.60) |
| Sterile blade or scissor available | 400/416 | 96.56 (94.42 to 97.90) | 98/99 | 99.00 (93.21 to 99.86) | 374/477 | 76.15 (70.79 to 80.79) | 98/109 | 88.55 (80.54 to 93.53) |
| Bag and mask available | 168/416 | 39.15 (34.16 to 44.37) | 76/99 | 76.12 (66.58 to 83.60) | 22/477 | 05.67 (03.63 to 08.77) | 18/109 | 18.73 (12.17 to 27.69) |
| Suction device available | 312/416 | 74.57 (69.54 to 79.02) | 80/99 | 80.17 (70.96 to 86.99) | 99/477 | 17.89 (14.57 to 21.76) | 60/109 | 58.06 (48.55 to 67.00) |
| Assistant birth attendant ready to intervene | 206/416 | 49.97 (44.69 to 55.24) | 76/99 | 76.13 (66.58 to 83.62) | 415/476 | 84.87 (80.55 to 88.37) | 95/109 | 88.14 (80.86 to 92.89) |
| Soon after birth (within 1 hour) | 415 | | 99 | | 477 | | 109 | |
| Clamps or ties cord correctly | 260/415 | 61.72 (56.40 to 66.77) | 56/99 | 58.42 (48.45 to 67.74) | 82/477 | 19.57 (15.33 to 24.57) | 32/109 | 32.46 (24.08 to 42.14) |
| Immediately dries the baby with clean towel | 412/415 | 99.23 (97.60 to 99.75) | 99/99 | 100 | 441/477 | 92.37 (89.52 to 94.49) | 103/109 | 93.96 (87.18 to 97.27) |
| Assesses for vaginal and perineal lacerations | 393/415 | 94.81 (92.11 to 96.63) | 95/99 | 96.18 (90.23 to 98.57) | 470/477 | 99.54 (96.91 to 99.31) | 102/109 | 93.13 (86.24 to 96.70) |
| Assesses for placenta completeness | 235/414 | 55.17 (49.83 to 60.40) | 45/99 | 47.42 (37.65 to 57.40) | 171/476 | 37.02 (32.05 to 42.28) | 33/109 | 33.18 (24.74 to 42.86) |
| Assesses newborn for need of reference | 413/415 | 99.71 (98.65 to 99.94) | 98/99 | 99.00 (93.21 to 99.86) | 418/477 | 86.24 (82.47 to 89.30) | 82/108 | 74.46 (65.12 to 81.99) |
| Takes mother vital signs within 15 min after | 208/415 | 51.82 (46.52 to 57.09) | 60/99 | 58.38 (48.26 to 67.85) | 234/477 | 47.01 (41.69 to 52.40) | 38/108 | 38.44 (29.52 to 48.21) |
| Palpates uterus within 15min after birth | 337/415 | 83.78 (79.75 to 87.14) | 79/99 | 79.44 (70.16 to 86.39) | 289/477 | 59.11 (53.44 to 64.55) | 45/108 | 44.80 (35.61 to 54.53) |
| Initiates breast feeding within 1 hour | 298/395 | 75.85 (71.01 to 80.06) | 63/93 | 67.36 (57.08 to 76.20) | 184/434 | 48.24 (42.46 to 54.06) | 28/100 | 30.14 (21.72 to 40.16) |
| Skin-to-skin contact initiated within 1 hour | 331/406 | 81.81 (76.98 to 85-81) | 74/97 | 76.02 (66.43 to 83.56) | 275/465 | 62.14 (56.94 to 67.09) | 34/103 | 35.09 (26.29 to 45.03) |
| Informs companion/woman to call if complications | 350/415 | 85.15 (80.92 to 88.57) | 95/99 | 95.80 (89.20 to 98.44) | 365/474 | 78.03 (73.47 to 82.00) | 74/105 | 70.86 (61.39 to 78.81) |
| Before discharge from the health facility | 411 | | 93 | | 474 | | 106 | |

Continued

**Figure 1** The box plots depict the overall scores of quality of care computed for each pause point by health facility type and by country. The maximum quality score for any pause point is equal to the number of essential best practices observed at that pause point. PHCF, primary healthcare facilities; RH, referral hospital.

### Compliance to EBPs before discharge from health facility

When leaving the facility, 47.14% (95% CI 37.09% to 47.36%) of women in Burkina Faso were offered a discussion on post partum family planning with a significantly higher performance in referral hospitals (p<0.001). In Cote d'Ivoire, the quality of breastfeeding was assessed in 26.48% (95% CI 21.46% to 32.19%) cases and was best adhered to in PHCFs (p<0.001).

The levels of adherence to all EBPs per pause point, country and health facility type are presented in table 2.

The analysis showed no significant variation between night and day shifts, nor there was a difference between weekdays and weekends in adherence to EBPs at any pause point.

The overall mean quality scores were consistently higher in referral hospitals than PHCFs in Burkina Faso at all pause points (p<0.01) except immediately post partum when there was no significant difference. In Côte d'Ivoire, the overall quality score was better in PHCFs at admission (p<0.001), immediately post partum (p<0.001), and before discharge from the health facility (p<0.01). It was higher in referral hospitals at the immediate prepushing phase (p<0.01). The distribution of the quality scores by pause point and by country is depicted in figure 1.

### DISCUSSION

In this study, we assessed healthcare providers' adherence to EBPs in delivery care using direct observation. Our results showed a wide diversity in adherence to EBPs ranging from very low to very high. Overall, adherence to EBPs tended to be better in referral hospitals as compared with PHCFs in Burkina Faso. In Côte d'Ivoire, the compliance was higher in PHCFs than referral hospitals. Important quality gaps were shown in both countries.

**Table 2** Continued

| Essential best practices | Burkina faso | | | | Côte d'Ivoire | | | |
| --- | --- | --- | --- | --- | --- | --- | --- | --- |
| | PHCF | | RH | | PHCF | | RH | |
| | n/N | % (95% CI) | n/N | % (95% CI) | n /N | % (95% CI) | n/N | % (95% CI) |
| Assesses the mother for vaginal bleeding | 291/408 | 70.82 (65.54 to 75.59) | 73/93 | 79.82 (70.59 to 86.69) | 310/474 | 68.68 (63.46 to 73.46) | 64/105 | 64.62 (55.10 to 73.10) |
| Assesses the baby for the quality of breast feeding | 227/408 | 55.56 (50.16 to 60.84) | 49/92 | 52.46 (42.14 to 62.59) | 129/474 | 26.60 (21.55 to 32.35) | 6/106 | 06.43 (02.92 to 13.58) |
| Discusses postpartum family planning with the woman/couple | 173/409 | 42.01 (36.94 to 47.26) | 64/92 | 67.24 (56.67 to 76.31) | 169/473 | 29.10 (24.52 to 34.12) | 35/106 | 30.19 (22.27 to 39.50) |
| Plans postpartum visits and informs to consult if complications | 349/409 | 84.28 (79.20 to 88.30) | 86/92 | 93.21 (85.60 to 96.95) | 380/474 | 77.68 (72.94 to 81.81) | 73/106 | 67.85 (58.20 to 76.18) |

*Per national guidelines of both countries, a partograph is indicated at admission only if cervix ≥4cm.
PHCF, Primary Healthcare Facility; RH, referral hospital.

Hand hygiene (washing hands before examination and wearing sterile gloves for each vaginal examination) practices were particularly very poor in both countries despite the widespread availability of hand hygiene facilities in both countries (hand washing facilities with soap and clean water were found in 94.74% and 74.96% of the maternity wards of surveyed healthcare facilities in Burkina Faso and Côte d'Ivoire, respectively). With the equipment available, adherence to hygiene routines is a fundamental and non-demanding practice for prevention of infections.[22 23] Yet, non-compliance is widespread and remains a source of concern globally.[24 25]

History-taking of critical information such as vaginal bleeding or blurred vision during pregnancy was suboptimal in Burkina Faso and poor in Côte d'Ivoire. Omission of blood pressure measurement was common, and occurred in one out of four women in Côte d'Ivoire. Temperature was not measured in two-thirds of the women in labour, missing an opportunity to identify ongoing episodes of infection.

These gaps in initial assessment can cause at-risk patients or prevailing complications to go undetected. It is also noteworthy that, despite being an ancient and long-systematised intervention, a partograph to monitor labour was used during less than half of the deliveries in both countries. The low utilisation of the partograph and its poor quality when used are well recognised and have been widely reported in the literature.[26–28]

Assessment of vaginal bleeding and perineal lacerations and preparation of a prefilled syringe with uterotonics for AMTSL were satisfactorily adhered to in both countries (>75%). These are important practices for the prevention, early detection, and management of postpartum haemorrhage (PPH), a leading cause of maternal death.[4] Bedside availability of prepared oxytocin improves its timely administration.[29] Increased compliance to these standards may be the result of rising awareness of the importance of PPH as a leading cause of death and the emphasis put on its prevention and management in both initial and refresher trainings for healthcare providers.

Implementation of other practices equally relevant to PPH: the performance of 'assessment of the completeness of the placenta' and 'monitoring of maternal vital signs' was, however, very low, and only complied with for around half of the women in Burkina Faso for both EBPs, and to one-third and three-sevenths of the women, respectively, in Côte d'Ivoire. These are life-saving low-cost practices, but require health-worker awareness and knowledge on the purpose and rationale for providing these services, in addition to clinical skills in executing them. The low proportion of women in both countries that received these interventions represent unmet needs of urgent action to address these cost-effective practices. Our findings also highlight the complexity of the peripartum and immediate postpartum care, wherein a series of equally important steps need to be taken to minimise risks of maternal morbidity, and how failure to address all of these puts women at risk of severe consequences.

Thermal management of newborns was satisfactorily adhered to in both countries through the availability of clean towels before the woman started pushing and drying the newborn immediately post partum. Compliance with other life-saving interventions for newborns (skin-to-skin contact, early initiation of breast feeding) was beyond 75% women in Burkina Faso and barely half of the women in Côte d'Ivoire.

Overall, supportive services (encouraging companion to assist in the delivery, identifying and preparing an assistant birth attendant ready to help) at any point in time tend to be less adhered to as compared with other standards of care.

The health facilities in both countries rarely used the opportunity to provide post-partum contraceptive advice to the patients. The same has been previously reported[20] and is suggestive of health workers' propensity to give precedence to interventions needed to prevent or treat imminent and tangible complications. Given the large unmet needs of modern contraceptives in both countries, implementation of postpartum family planning programmes is an important strategy to close the gap. Knowledge translation and follow-up of patients is needed between healthcare workers engaged in intrapartum care and family planning.

Noticeably, a significant proportion of women and or companions received information related to potential complications and signs of danger and what to do in case they occur in both countries from admission to discharge. Apart from the immediate postpartum period wherein there was no significant difference, compliance with standards in Burkina Faso tended to be better at all junctures in referral hospitals than PHCFs. Such a systematic difference was not observed in Côte d'Ivoire. This difference may be because in Burkina Faso, there is a wider existence of categories of auxiliary nurses and auxiliary midwives who predominantly work in PHCFs, while in Côte d'Ivoire, deliveries are attended to by nurses and midwives at PHCFs as well as referral hospitals.

### Study limitations

In interpreting our results, some limitations should be considered. We carried out a direct observation of facility-based deliveries in a sample of health facilities in two selected regions and the findings may not be necessarily representative of each individual country. Although we did not provide information to healthcare providers on the items being observed, we cannot rule out the Hawthorne effect and some may have changed their healthcare practices during the study period. The data collection was carried out by eight data collectors in each country. Although they were trained by the same research team to standardise processes and judgements, we cannot rule out issues related to inter-rater reliability. Further analysis including the level of adherence to EBPs according to providers qualification and working experience would have added value to this work. Finally, the non-compliance to a given standard does not always imply

that the observed healthcare provider failed to do so. In some cases, the supplies needed to administer the intervention may have been unavailable.

## CONCLUSION

Overall, compliance to EBPs during delivery was insufficient and important gaps in quality of care were identified. A significant number of women in Burkina Faso and Cote d'Ivoire missed out on important interventions, especially those related to infection prevention and control and prevention of PPH. In addition, there were lost opportunities in relation to implementation of postpartum contraceptive strategies. Conditional on the availability of supplies, the investigated EBPs were relatively low cost and easy to implement. Quality-improvement interventions in the form of reminder tools for healthcare providers, such as the WHO-SSC, have the potential to sustain currently performed best practices while filling the gaps that persist.

**Acknowledgements** We are grateful to WHO Human Reproduction Program (WHO/HRP) for funding the research.

**Contributors** All authors meet the criteria recommended by the International Committee of Medical Journal Editors (ICMJE). All authors made substantial contributions to conception and design, acquisition of data or analysis and interpretation of data. SK,TM and MLA-Y conceived the study design. TM analysed the data and proposed the first draft of the paper. SK, MLA-Y, RKK, WMEY, AT-K, FBS, AS-C and AT contributed to interpret the data and have revised and contributed to the successive drafts. The final version of the paper was revised and approved by all the authors.

**Funding** This work was supported by WHO/HRP.

**Competing interests** None declared.

**Patient consent for publication** Not required.

**Ethics approval** The research protocol obtained ethical approvals from the national ethics review committees of Burkina Faso (reference number 2017-4-043) and Côte d'Ivoire (161–18/MSHP/CNE SVS-Kp). It was also approved by the ethics review committee of the WHO (protocol ID ERC.0002951).

**Provenance and peer review** Not commissioned; externally peer reviewed.

**Data availability statement** Data are available on reasonable request. The data presented in this study are from the Checklist study conducted in Burkina Faso and Côte d'Ivoire. Any request to access the data can be sent to the principal investigatior of the study, SK at senikouanda@gmail.com. Access will be granted only after careful and due consideration of the compliance with the ethics requirements and the data policy of the Institut de Recherche en Sciences de la Santé.

**ORCID iD**
Tieba Millogo http://orcid.org/0000-0003-3548-3789

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
