## [Reviewer comments · BMJ Open]

ARTICLE DETAILS

TITLE (PROVISIONAL)	Quality of maternal and newborn care in limited-resource settings: a facility-based cross-sectional study in Burkina Faso and Côte d'Ivoire
AUTHORS	Millogo, Tieba; Agbre-Yace, Marie; Kourouma, Raissa K; Yaméogo, W. Maurice E.; Tano-Kamelan, Akoua; Sissoko, Fatou; Soltié-Coulibaly, Aminata; Thorson, Anna; Kouanda, Seni

VERSION 1 – REVIEW

REVIEWER	Ribka Amsalu Save the Children USA
REVIEW RETURNED	30-Dec-2019

GENERAL COMMENTS	Important study that will contribute to improvement in clinical practice. Comments Abstract. Avoid the term "sub-optimal", indeed as you describe in the results section adherence to EBP is varied by practice from <10% to >90% and it is unhelpful to generalize. Include 95% CI for each proportion in the results section of abstract and throughout paper. Suggest - to not compare BF with CID, the selected HFs as you describe in the method section are not representative of each country. Data analysis and results, lacks depth.  1. Did the researchers compare EBP at the hospital versus PHCFs per country? that could be more informative for action than comparing the countries. Similarly, variation by PHCFs in each country could be informative for public health action 2. Was there variation in adherence to EBP - night shift versus day shift? was the observation done 24/7? 3. Level of experience of the health care worker. was data collected on provider (MD, nurse, midwife, obs/gyn...) and their year of experience or recent training? could that influence who adhered and did not adhere to EBP? that is actionable information. 5. Box plot - misleading. Each pause point has varied number of EBPs. Having a box plot by number of EBPs done in one plot is misleading. Standardize the unit of measurement as out of 100 or something similar. 6. Study limitation - findings are NOT representative of each country or district of the country. 7. Add 95%CI for each proportion. There is a way to do it for cross-sectional studies.
--

	8. Number of birth events observed decreases from admission to discharge in all locations. What is the explanation? refusal to be observed? observer absent? or ? add an explanation and include as limitation of the study 9. Table 3. I did not do the calculations, but did you change the denominator for each pause point? If Table 2 is the total observed mother-baby pair at each pause point then your denominator for Table 2 - will have missed data points i.e. not yes/no but rather unobserved. 10. What is partograph use when "indicated" mean? partograph is universal unless the client arrived at 2nd stage of labor, if that is the case the woman wouldn't be in your sample, correct? Minor comments Proof reading is advised "newborn" not "new born" "bag and mask " not "face mask" "wearing sterile gloves for each vaginal examination". is the low % observed due to use of unsterile glove or no gloves at all. This is an important distinction.
--	--

REVIEWER	Bareng A.S Nonyane Johns Hopkins Bloomberg School of Public Health
REVIEW RETURNED	22-Jan-2020

GENERAL COMMENTS	This is an important study in this setting where maternal and child mortality rates are still unacceptably high. It is thus a relevant contribution to the literature that could inform policies for improving the quality of maternal and newborn care. It is a cross-sectional study involving direct observation of providers' management of deliveries and discharge. Setting: Burkina Faso and Cote d'Ivoire facilities intra-partum and post-partum care, observations of practice Indicators: Adherence to Essential Best Practices; overall quality score Most methods are adequately described, except those mentioned below. The authors conclude that best practices were not adhered to and they identified quality of care gaps in the procedures Major comments:  • How were the 5 districts selected? 'Randomly' is not enough to describe the approach used. What criteria were used? • Was there a training session or a subset of observations that could be used to test inter-rater reliability of observers? • Given that observations were clustered in clinics that had varying numbers of deliveries, how would these results differ if they were weighted by clinic size. The authors should run these calculations and comment on the effect of weighing. • Discussion: authors allude to 'widespread availability of hand hygiene facilities ' but they note that the results are not presented. This is surprising. The reader should be able to understand the context fully. The availability of supplies is also alluded to in the conclusion. So one assumes that is a very important finding. What facilities are available? What may be the reasons for the clinicians not using them?
---

	 • Was the t-test, rather than the non-parametric counterpart, appropriate for comparing the mean quality scores between the two countries? I'd say maybe the latter is better. • In study setting, the authors mention the maternal mortality ratios of the two countries but they do not say what they are. CAN they say the specific numbers? Other comments:  • The manuscript needs English copy-editing throughout. • Minor typos. Example missing "to" before "healthcare" in the sentence "Non-disclosing of the items being observed the healthcare providers whose care practices were being observed;
--	---

VERSION 1 – AUTHOR RESPONSE

Reviewer 1	
COMMENTS	RESPONSES
Important study that will contribute to improvement in clinical practice.	Thank you.
Comments 1. Abstract. Avoid the term "sub-optimal", indeed as you describe in the results section adherence to EBP is varied by practice from <10% to >90% and it is unhelpful to generalize. Include 95% CI for each proportion in the results section of abstract and throughout paper. Suggest - to not compare BF with CID, the selected HFs as you describe in the method section are not representative of each country.	The term sub-optimal was removed from the title and we have re-analysed the data using weights and are now reporting all results along with the 95% CI intervals. Instead of comparing the two countries we are now comparing PHCFs vs RH within each country. Thank you for this comment. See table of results page 14-15
2. Data analysis and results, lacks depth. Did the researchers compare EBP at the hospital versus PHCFs per country? that could be more informative for action than comparing the countries. Similarly, variation by PHCFs in each country could be informative for public health action.	Ok. We have re-analyzed the data to take this into account. We are now reporting the adherence to EBPs for each pause-point comparing PHCFs and hospitals in each country.
3. Was there variation in adherence to EBP - night shift versus day shift? was the observation done 24/7?	Yes, the observations were carried out 24/7. The analysis showed no significant variation between night and day shifts, nor there was a difference between weekdays and weekends. Thanks
4. Level of experience of the health care worker. was data collected on provider (MD, nurse, midwife, obs/gyn...) and their year of experience or recent training? could that influence who adhered and did not adhere to EBP? that is actionable information.	We agree with the reviewer that this would have been more informative. Notwithstanding this absence, we believe the aim of reporting on the level of adherence to EBPs was achieved. Thanks

5. Box plot - misleading. Each pause point has varied number of EBPs. Having a box plot by number of EBPs done in one plot is misleading. Standardize the unit of measurement as out of 100 or something similar.	Ok noted. We have now plotted separate graphs for each pause point to account for this. See figure 1
6. Study limitation - findings are NOT representative of each country or district of the country.	Ok noted. We have included this limitation (see strengths and limitations page 3). Thanks
7. Add 95%CI for each proportion. There is a way to do it for cross-sectional studies.	Ok. Correction done. Thanks
8. Number of birth events observed decreases from admission to discharge in all locations. What is the explanation? refusal to be observed? observer absent? or? add an explanation and include as limitation of the study.	The decrease in the numbers from one pause point to another was due to either reference/ transfer of the mother-newborn pair to a higher level of care or to the observer failing to observe at a given pause point (shifts between observers). The other reason why a mother-newborn pair could also not be observed at the fourth pause-point was them still being in the health facility at the time the data collectors completed their overall stay in that particular health facility. We did not collect information on the reason why a woman was not observed at a specific pause-point. Thus, we do not have the share of the missingness between those reasons. The reference to higher level of care may be related to poor quality of care leading to complications. However, we do not anticipate the same with the failure to observe at a specific pause point that can reasonably be considered as resulting in missingness completely at random. Given the small overall amount of decrease (<5%) between successive pause points, we don't see it as a major subject of concern. Explanation was added to the text (see page 11 line 30-38). Thanks
9. Table 3. I did not do the calculations, but did you change the denominator for each pause point? If Table 2 is the total observed mother-baby pair at each pause point, then your denominator for Table 2 - will have missed data points i.e. not yes/no but rather unobserved.	Yes. The denominators change from one pause point to another and within the same pause point there could be a change from one item to another. Please see table 2 page14 for more clarity. Thanks
10. What is partograph use when "indicated" mean? partograph is universal unless the client	Per national guidelines of both countries, a partograph is indicated at admission only if cervix \geq 4 cm. Women that were admitted with

arrived at 2nd stage of labor, if that is the case the woman wouldn't be in your sample, correct?	cervix<4cm were not eligible to start a partograph at the admission. And partograph use at the admission was considered not indicated.
11. Proof reading is advised "newborn" not "new born" "bag and mask " not "face mask" "wearing sterile gloves for each vaginal examination". is the low % observed due to use of unsterile glove or no gloves at all. This is an important distinction.	Ok thanks.
Reviewer 2	
COMMENTS	RESPONSES
1. This is an important study in this setting where maternal and child mortality rates are still unacceptably high. It is thus a relevant contribution to the literature that could inform policies for improving the quality of maternal and newborn care. It is a cross-sectional study involving direct observation of providers' management of deliveries and discharge.	Thank you.
2. How were the 5 districts selected? 'Randomly' is not enough to describe the approach used. What criteria were used?	The two regions were purposively selected based on the presence of a geographic location of the research centers that partner to conduct the study. In each region we randomly drew five from the existing (6 in each country) without further criteria.
3. Was there a training session or a subset of observations that could be used to test inter-rater reliability of observers?	Yes. During the training session a one-day field test was organized where the same pause point was observed by two data collectors along with the supervisor (member of the research team). The results were compared to that of the supervisor and the discrepancies were discussed. We have added some more text on that in the manuscript page 7.
4. Given that observations were clustered in clinics that had varying numbers of deliveries, how would these results differ if they were weighted by clinic size. The authors should run these calculations and comment on the effect of weighing.	Ok. Noted with thanks. We have re-analyzed the data to account for this. We have re-analyzed the data to take this into account. See table of results page 14-15
5. Discussion: authors allude to 'widespread availability of hand hygiene facilities 'but they note that the results are not presented. This is surprising. The reader should be able to understand the context fully. The availability	The facilities alluded to in this section are related to hands hygiene facilities. The research findings showed that 94,74% and 74,96 % of health care facilities had a hand

of supplies is also alluded to in the conclusion. So one assumes that is a very important finding. What facilities are available? What may be the reasons for the clinicians not using them?	washing facility with soap in the maternity ward respectively in Burkina Faso and Côte d'Ivoire. this information was added to the manuscript. Page 16, line 115-117
6. Was the t-test, rather than the non-parametric counterpart, appropriate for comparing the mean quality scores between the two countries? I'd say maybe the latter is better.	We have compared the quality scores between facility types using a Wilcoxon Ranksum test.
7. In study setting, the authors mention the maternal mortality ratios of the two countries, but they do not say what they are. CAN they say the specific numbers?	Ok. Figures included, Page 5 line 112_113. Thanks
8. Other comments:  • The manuscript needs English copy-editing throughout. • Minor typos. Example missing "to" before 'healthcare" in the sentence "Non-disclosing of the items being observed the healthcare providers whose care practices were being observed; 	Ok noted. The revised version was submitted to copyediting service.